# Adrenocortical Carcinoma and CT Assessment of Therapy Response: The Value of Combining Multiple Criteria

**DOI:** 10.3390/cancers12061395

**Published:** 2020-05-28

**Authors:** Roberta Ambrosini, Maria Carolina Balli, Marta Laganà, Martina Bertuletti, Luca Bottoni, Filippo Vaccher, Deborah Cosentini, Marco Di Terlizzi, Sandra Sigala, Salvatore Grisanti, Guido Alberto Massimo Tiberio, Alfredo Berruti, Luigi Grazioli

**Affiliations:** 1I Radiology Unit, ASST Spedali Civili, 25123 Brescia, Italy; m.balli@unibs.it (M.C.B.); m.bertuletti@unibs.it (M.B.); l.bottoni@unibs.it (L.B.); f.vaccher@unibs.it (F.V.); mditer@gmail.com (M.D.T.); luigi.grazioli@asst-spedalicivili.it (L.G.); 2Medical Oncology Unit, Department of Medical and Surgical Specialties, Radiological Sciences and Public Health, University of Brescia at ASST Spedali Civili, 25123 Brescia, Italy; m.lagana001@unibs.it (M.L.); d.cosentini@unibs.it (D.C.); salvatore.grisanti@unibs.it (S.G.); alfredo.berruti@unibs.it (A.B.); 3Section of Pharmacology, Department of Molecular and Translational Medicine, University of Brescia, 25123 Brescia, Italy; sandra.sigala@unibs.it; 4Surgical Unit, Department of Medical and Surgical Specialties, Radiological Sciences, and Public Health, University of Brescia, ASST Spedali Civili di Brescia, 25123 Brescia, Italy; guido.tiberio@unibs.it

**Keywords:** adrenocortical carcinoma, ACC, computed tomography, CT, response criteria, RECIST 1.1, Choi, tumor volume

## Abstract

We evaluated tumor response at Computed Tomography (CT) according to three radiologic criteria: RECIST 1.1, CHOI and tumor volume in 34 patients with metastatic adrenocortical carcinoma (ACC) submitted to standard chemotherapy. These three criteria agreed in defining partial response, stable or progressive disease in 24 patients (70.5%). Partial response (PR) was observed in 29.4%, 29.4% and 41.2% of patients according to RECIST 1.1, CHOI and tumor volume, respectively. It was associated with a favorable prognosis, regardless of the criterion adopted. The concordance of all the 3 criteria in defining the disease response identified 8 patients (23.5%) which displayed a very good prognosis: median progression free survival (PFS) and overall survival (OS) 14.9 and 37.7 months, respectively. Seven patients (20.6%) with PR assessed by one or two criteria, however, still had a better prognosis than non-responding patients, both in terms of PFS: median 12.3 versus 9.9 months and OS: 21 versus 12.2, respectively. In conclusions, the CT assessment of disease response of ACC patients to chemotherapy with 3 different criteria is feasible and allows the identification of a patient subset with a more favorable outcome. PR with at least one criterion can be useful to early identify patients that deserve continuing the therapy.

## 1. Introduction

Adrenocortical carcinoma (ACC) is a rare neoplasm, with an estimated incidence in Western countries between 0.7 to 2 cases per million population per year [1]. With the increasing accuracy of diagnostic modalities and their more frequent use, nearly 15% of ACC cases are incidentally diagnosed and the risk of malignancy in incidental lesions has been related to tumor size (sensitivity and specificity values of 97% and 52% for lesions > 4 cm and 91% and 80% for tumors > 6 cm, respectively) [2]. In patients with adrenal tumors of 1–4 cm both the European Guidelines for management of adrenal incidentalomas [3] and the American College of Radiology Paper on the management of incidental adrenal masses [4] underline the role of unenhanced CT to detect microscopic fat content in solid adrenal lesions, with an attenuation threshold of 10HU, which is considered fundamental (with sensitivity and specificity values of 71% and 98%, respectively) in the differential diagnosis between benign and malignant lesions [5]. ACC, however, is frequently diagnosed in advanced stages: 18–26% of patients are in stages III and 21–46% in stage IV at presentation [6]. The respective five-year survival is 24–50% and 0–17% for stage III and IV patients, respectively [7,8]. Cortisol hypersecretion [9], complete surgical resection and proliferative activity, assessed by Ki67 expression [10] are additional independent prognostic parameters.

The etoposide, doxorubicin and cisplatin combination regimen, administered in association with oral mitotane (EDP-M), is the mainstay of therapy for advanced/metastatic ACC [11,12]. According to the ENSAT guidelines [13], contrast-enhanced CT (CECT) of the chest and abdomen/pelvis is the reference imaging method, both for tumor staging and disease monitoring during therapy, while MRI of the abdomen and FDG-PET could offer additional information in selected cases. Response Evaluation Criteria in Solid Tumors (RECIST 1.1) [14] is the reference system for response evaluation of solid tumors to systemic antineoplastic treatments and is currently employed in the evaluation of the activity of chemotherapy in advanced ACC. RECIST 1.1 is based on detection of changes in tumor size, measured as the sum of the two longest axial diameters. The relationship between change in diameter and volume of tumors is based on the assumption that solid neoplastic lesions are spherical and, consequently, that proportional changes of tumor volume and product of perpendicular dimensions follow changes in tumor diameter, and vice versa. In practice, however, tumors commonly show odd shapes and not all parts equally respond to treatment. Another limitation of RECIST 1.1 is that it relies on the presumed correlation between tumor volume burden and planar dimensions. The assessment of tumor response according to volumetric criteria could be more advantageous than RECIST for assessing response to treatment in cancer patients, better showing size changes even in large, irregular lesions [15,16]. Moreover, not infrequently the tumor changes after treatment may be characterized by an increase in the necrotic component, which can be associated with stabilization or even an increase of the tumor size. The post-chemotherapy changes of tumor structure may present with the appearance of necrotic/hypo enhancing areas and these effects can be assessed as attenuation changes, measured in Hounsfield units (HUs) at contrast-enhanced CT scan. This is the basis for evaluation of response using the Choi criteria. These criteria were repeatedly found to be more helpful than RECIST to define advanced/metastatic cancer patients who benefit from target therapies such as imatinib [17] or sunitinib [18]. The introduction of multidetector-row CT (MDCT) scanners, in the clinical radiology practice, allows isotropic acquisition of extensive anatomic data, with postprocessing analysis, tumor segmentation, evaluation of tumor volume burden and its variations during therapies. These advances in imaging technology allow not only the measurement of bidimensional tumor size, but also different postprocessing analysis such as semiautomatic outlining of tumor boundaries (segmentation) and assessment of different tumor characteristics such as accurate definition of lesion volume, attenuation changes, favoring the detection of tumor response using multiple response criteria.

In the present study, we prospectively evaluated disease response of advanced/metastatic ACC patients, uniformly treated with the standard EDP-M regimen, using RECIST 1.1, Choi and volume criteria. The study aimed to correlate response assessment with each criterion with progression free survival (PFS) and overall survival (OS) and to evaluate whether the combination of the three criteria could offer more helpful information on patients’ outcome.

## 2. Results

Demography and characteristics of the 34 patients included in the study are shown in Table 1. They were 24 females and 10 males, median age at diagnosis was 46.3 years (range 16–71). Sixteen patients (47.1%) did not have distant metastases and were initially addressed to surgery, while the remaining 18 patients (52.9%) had unresectable locally advanced or metastatic disease. Eighteen patients (52.9%) had secretory tumor. According to mENSAT ACC staging system [6,13,19], 4 patients (11.8%), were at stage III, 16 patients at stage IV A (47.1%), 6 at stage IV B (17.6%) and 8 at stage IV C (23.5%), before EDP-M administration, respectively. The distribution of the analyzed target lesions was as follows: 24 primary adrenal tumors (70.6%), 4 loco-regional recurrent solid lesions (11.8%), 2 liver metastases (5.9%), 2 mediastinal lymph-nodes (5.9%) and 2 peritoneal localization (5.9%).

At the time of the analysis 20 patients (58.8%) were dead.

In the present series no advanced ACC patients attained a complete response to the therapy with any adopted criteria. According to RECIST 1.1, a partial response (PR) was obtained in 10 patients (29.4%), 18 patients (52.9%) had stable disease (SD) and 6 patients (17.6%) showed disease progression (PD). As regard as Choi criteria, 14 patients (41.2%) were classified as PR, 13 patients (38.2%) had SD and 7 patients (20.6%) had PD. According to volumetric criteria, 10 patients (29.4%) obtained a PR, 17 patients (50%) SD and 7 patients (20.6%) PD. Overall, in 24 patients (70.5%) there was an agreement of all the three criteria in defining PR, SD or PD. Concerning the PR definition, this was recognized by all the three criteria in 8 patients (23.5%). Among discordant results, 2 patients (5.9%) considered responsive using RECIST 1.1 were not recognized as such using Choi criteria: 1 patient (2.9%) and volumetric criteria: 1 patient (2.9%). Four patients (11.7%) considered responsive applying the Choi criteria were classified as unresponsive by both RECIST 1.1 and volumetric criteria; 2 further patients (5.9%) classified as responsive according to the Choi criteria were not confirmed by RECIST 1.1 (1 patient-2.9%) and volume criteria (1 patient-2.9%).

Finally, 10 patients (29.4%) considered responsive by the volumetric criteria were not confirmed by Choi in 1 patient (2.9%) and RECIST 1.1 in another patient (2.9%). An example of a large pelvic secondary implant, considered as a PR using the Choi criteria, otherwise classified as PD when applying RECIST 1.1 and volumetric criteria is shown in Figure 1.

Overall, PR recognized by at least one criterion was observed in 15 patients (44.1%).

We analyzed the correlation among each response criterion and patient outcome, in term of PFS (Table 2) and Os (Table 3). Responding patients, assessed with the RECIST 1.1 criteria, attained a higher OS (median 37.7 months) with respect to patients with SD (18.7 months) or PD (median 14.3 months). The corresponding median PFS was 14.9, 9.9 and 1.8 months, respectively.

As regard as Choi, median OS was 25.4 months in responding patients, 18.7 months and 14.3 months in those with SD or PD, respectively. Median PFS in the 3 groups was 14.9, 9.9 and 10.8 months, respectively.

With respect to Volume criteria, median OS was 37.7 months in responding patients and 18.7 and 14.3 months in those obtaining SD or PD. The corresponding median PFS was 14.9, 9.9 and 2.6 months, respectively.

The Receiver operating characteristic curve (ROC) analysis revealed that all the three response criteria provided a similar moderate accuracy for predicting either PFS (area under curve [AUC] being 0.595, 0.668 and 0.654 for RECIST, CHOI and Volume criteria, respectively) or OS (AUCs: 0.717, 0.704 and 0.717, respectively).

We further explored the additional information that Choi and volume response added to the RECIST response and observed that median OS was 37.7 months in patients with RECIST response, 21 months in patients obtaining Choi response, but no response according to RECIST and 12.2 months in non-responders. The corresponding median PFS values were 14.9, 12.3 and 9.9 months, respectively. Patients who obtained a response with volume criteria, but not with RECIST had a median OS of 19.2 months and PFS of 15.3 months as opposed with not responding patients in which median OS and PFS were 14.3 and 9.9 months, respectively.

Moreover, we analyzed Overall Survival (OS) and Progression Free Survival (PFS) in patients attaining PR by at least one criterion versus others (SD and PD). The median OS was 25.4 months (19.4–31.4) in patients with PR and was 12.2 months (8.2–16.2) in the non-responding group. As regard as PFS, the median was 14.9 months (11.8–18) in PR patients and 9.9 months (6.8–12.9) in patients with SD or PD (Figure 2 and Figure 3).

Finally, we analyzed OS and PFS stratifying our patients in three groups, as follows: (A) patients in whom the response assessed with all the three criteria were concordant; (B) responding patients according to one or two criteria, but not all the three criteria; (C) patients in whom there was no response with any of the adopted criteria.

In the group A, the median OS was 37.7 months (95% CI: 19.8–55.6), versus 21 months (95% CI: 17.3–24.6) in the group B and 12.2 months (95% CI: 8.2–16.2) in the patients of the group C (SD or PD), *p* = 0.005 (Figure 4).

Group A patients showed a better PFS: 14.9 months (95% CI: 12.5–17.2), than group B: median of 12.3 months (95% CI: 4.7–19.8) and group C: median 9.9 months (95% CI: 6.8–12.9), *p* = 0.107 (Figure 5).

## 3. Discussion

Imaging has a pivotal role in objectively defining tumor response or progression of cancer patients during systemic therapy. RECIST criteria, introduced in 2000 [20], are the most standardized, scientifically accepted and currently used system for tumor response evaluation. As mentioned in the introduction, these criteria, even after their revision [14], have several limitations linked to the heterogeneity of the forms and contours of tumor lesions and among different lesions in the same patient. These limitations can be addressed at least in part by the concomitant use of others response criteria. In patients affected by GIST, treated with the tyrosine kinase inhibitor imatinib, Choi criteria demonstrated that the degree of contrast enhancement at CT reflects vascular and interstitial volumes of the tumor, providing information about its structure and biological behavior, even in the absence of size variations [21,22]. In addition, the introduction in clinical radiology of dedicated software allows a fast and semiautomatic segmentation of the analyzed lesions with the automatic measurements of diameters as well as volume and attenuation changes. This approach aims to reduce the subjectivity of the analysis and to detect earlier subtle variations during and after systemic antineoplastic therapy, as well as to shorten the time spent by the radiologist for the manual tumor analysis, while increasing the reproducibility of the evaluation (Figure 1).

In this study, we have prospectively evaluated disease response at CT imaging of advanced/metastatic ACC patients submitted to EDP-M regimen, adopting RECIST 1.1, Choi and volumetric criteria (Figure 6). Partial response assessed by each criterium significantly correlated with patient outcome, both in terms of PFS and OS. However, the concordance among these criteria in defining the disease partial response was observed in 8 patients (23.5%), while in 7 patients (20.5%) the response observed with one criterion was not confirmed by one or 2 other criteria. We therefore adopted a comprehensive approach, considering as responders all patients showing a disease response with at least one criterion. Using this definition, the proportion of responders increased from 23.5% to 44% with respect to standard RECIST 1.1, and responding patients maintained a better PFS and OS than patients with SD and PD. We subsequently evaluated the prognostic effect of the response recognized by all three criteria than discordant response between criteria and disease stabilization or progression. The results showed that patients in whom the response was agreed upon by the three criteria obtained the best prognosis in terms of PFS and OS. Patients, whose response was identified by one or two criteria, but not by all the three obtained a better survival, although not a significant advantage in terms of PFS than patients with SD and PD.

These results have potential clinical implications. First, in the clinical course of patients undergoing EDP-M therapy we need to early identify which of them should stop a toxic treatment due to inefficacy with respect to those who deserve to continue the treatment. A disease response identified by at least one parameter could be useful in this respect, particularly in cases of discordant results. Noteworthy, the disease control in ACC patients submitted to EDP-M is subordinated not only to the cytotoxic effect of chemotherapy, but also to the achievement of mitotane levels in the therapeutic range, which usually occurs after 2–3 months from the beginning of therapy [23,24]. We have learned from the routine clinical practice that early progression of ACC patients to EDP-M according to RECIST 1.1 does not always mean treatment inefficacy [23,24]. The combination of the three methods of analysis could be pivotal, to identify true progressing patients from patients in which the treatment should be continued, despite initial disease progression.

Perhaps more important, surgery of residual disease is often considered in many patients with ACC who achieve an objective response to cytotoxic treatment. The long-term efficacy of the surgical approach of metastatic malignant disease is known to be higher in patients with an indolent disease or in those in which the disease is made inactive by treatment. The finding that patients in whom all the three criteria confirmed the response to therapy had the best prognosis is relevant since it could identify a patient subgroup with less aggressive disease that could potentially obtain benefit from adjunctive surgery if deemed feasible.

This study has several limitations, first of all it is clearly underpowered, only 34 patients were evaluated. Considering the 29% response rate obtained with the RECIST criteria alone and the 44% response obtained considering at least one of the 3 criteria, the study has a power of 0.36 with a one-tailed test (0.25 with a two tailed test) to demonstrate the observed absolute difference of 15% with an alpha error of 5%. Moreover, the few patients enrolled, the absence of a validation set and the different target lesions analyzed, with 24 primary tumors (70.6%), 4 loco-regional recurrent solid lesions (11.8%), 2 liver metastases (5.9%), 2 mediastinal lymph-nodes (5.9%) and 2 peritoneal localization (5.9%) are additional drawbacks of this study.

These limitations notwithstanding, our results underline the importance to combine these three response criteria to better define the prognostic role of disease response to therapy. The progressive implementation and development of advanced software for automatic analysis is crucial to achieve an accurate, objective analysis, together with a significant reduction of the radiologist’s time spent in the segmentation process.

## 4. Patients and Methods

In this prospective, observational, monocentric study we analyzed the clinical data and CT examinations of 58 locally advanced or metastatic ACC patients, treated in the Department of Oncology at ASST Spedali Civili of Brescia from November 2013 to September 2019. Among them, 24 were excluded due to incomplete/inadequate CT data (because CT examinations performed in different sites and with different imaging protocols were not always recognized from the IntelliSpace Portal Software used for the analysis), different anatomic districts (i.e., patients with exclusive lung metastases, where the semi-automatic segmentation process could be less accurate) and sizes of the lesions (very small lesions, not always recognized by the software), and/or inclusion in EDP-M chemotherapy regimen, therefore our final population consisted in 34 patients (Figure 7).

EDP-M was administered according to the following scheme: etoposide 100 mg/m^2^ (day 2–3–4), doxorubicin 20 mg/m^2^ (day 1), cisplatin 40 mg/m^2^ (day 3–4). Oral mitotane was administered concomitantly with chemotherapy at the starting dose of 1500 mg daily, with further progressive dose increments up to the maximum tolerated dose. Serum mitotane was monitored every 4 weeks and when the patients attained the therapeutic range the mitotane dose was tapered to maintain serum concentration within 14 and 20 mg/L. All patients were studied at the baseline (before starting of chemotherapy) and every 3 months, according to the ENSAT guidelines.

Multi Detector Computed Tomography (MDCT) examinations were carried out using two different scanners, Toshiba Aquilion (with 80 detectors array) and Philips Brilliance (with 64 detector arrays), before and after i.v. automatic injection of iodinated contrast agent (370 mgI/mL@3 mL/s-1.3 mL/Kg body-weight), using a bolus tracking technique. CT images of the chest and abdomen were acquired during the late arterial and venous phases after contrast injection. Image analysis was performed using a dedicated Software (Multi-Modality Tumor Tracking, IntelliSpace Philips Portal, version 10, Philips Healthcare, Best, The Netherlands) and focused to the analysis of solid lesions, as follows: primary adrenal tumors, local solid recurrence, recurrent peritoneal lesions, hepatic metastases and solid pulmonary lesions.

Two experienced Radiologists (R.A., M.D.T.) performed 3D semiautomatic segmentation of target lesions on the baseline examination and on the first CT after chemotherapy, using a dedicated software (Multi-Modality Tumor Tracking, IntelliSpace Philips Portal, version 10, Philips Healthcare, Best, The Netherlands). After segmentation, the software automatically compares the examinations in terms of the long and short axis, lesion volume and CT attenuation. Furthermore, aside from the calculation of the percentages of variations, the software provides a chart where relative changes of the analyzed tumor characteristics between multiple examinations are immediately evident. In this study, we evaluated tumor response considering the sum of the longest diameter of all lesions (RECIST 1.1 criteria), the presence of attenuation changes at CT (CHOI) and volume changes after therapies (Figure 8).

The assessment of response according to the RECIST 1.1, Choi and volumetric criteria are summarized in Table 4.

Study population characteristics were described using classical descriptive statistics: percentage, means and standard derivation, median and extreme values. PFS was defined as the time elapsing from the beginning of the EDP-M treatment until disease progression or death. Non-progressing patients still alive were censored at the last follow-up examination. OS was defined as the time interval between the date of EDP-M treatment start and the date of death from any cause or the last known alive date. The present study aimed to obtain exploratory information on the prognostic role of disease response evaluated with 3 different criteria—and the combination of them—in patients with advanced ACC submitted to systemic chemotherapy. Due to the exploratory nature of this study, a sample size was not calculated. All survival curves were calculated by the Kaplan–Meier method and differences compared by the log-rank test. Cox’s proportional hazards regression model was employed to assess the Hazard ratios (HRs) and 95% confidence intervals (95% CIs). Missing data were dealt with by excluding patients from particular analyses if their files did not contain data for the required variables. A Receiver operating characteristic curve (ROC) analysis was conducted to assess the global accuracy of disease response by the 3 criteria for predicting PFS and OS categorized at the median value. SPSS statistical software (version 23.00, Chicago, IL, USA) was used for statistical analysis. A *p* value of 0.05 was considered statistically significant.

## 5. Conclusions

The concomitant use of RECIST 1.1, volumetric and CHOI criteria in the assessment of disease response at contrast-enhanced MDCT, in advanced/metastatic ACC patients submitted to EDP-M chemotherapy, is feasible. Dedicated software and automatic procedures make it more objective and not excessively time-consuming for the radiologist. The concordance of all the three response criteria allows more accurate detection of a patient subset destined to obtain a good outcome, in which surgery of the residual disease could be justifiable, if technically feasible. The partial non-overlap between the response criteria allows instead to enlarge the number of patients deemed responsive to therapy and this can be useful in discriminating early after a few cycles, the patient for whom it is appropriate to continue therapy compared to those in whom it is more appropriate to stop treatment.

## Figures and Tables

**Figure 1 cancers-12-01395-f001:**
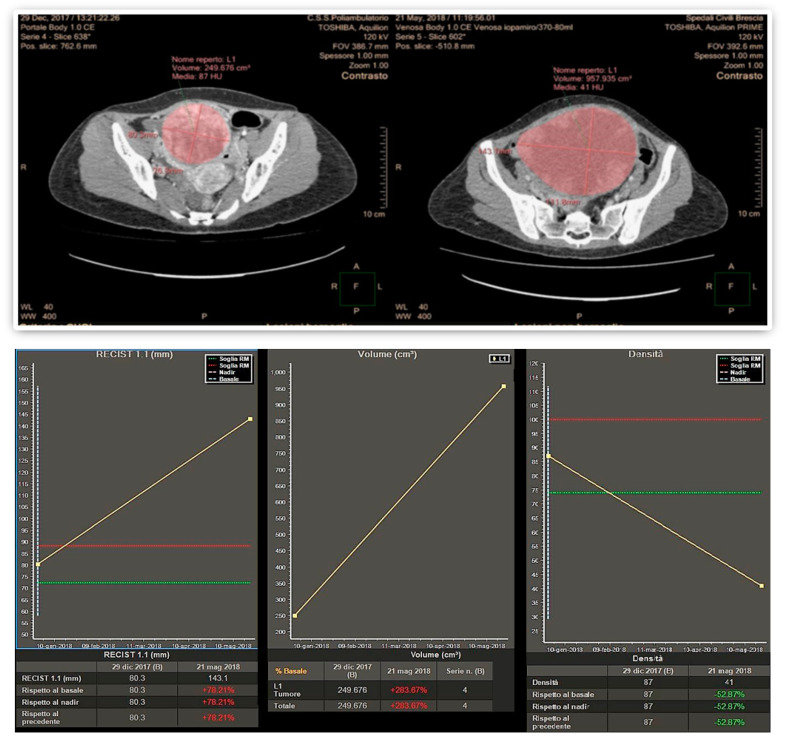
Disagreement in response assessment within the three criteria: according to Choi, the decrease in tumor attenuation was evaluated as a partial response, while according to both Volume and RECIST 1.1 its increase in planar dimensions and volume resulted in Progressive Disease.

**Figure 2 cancers-12-01395-f002:**
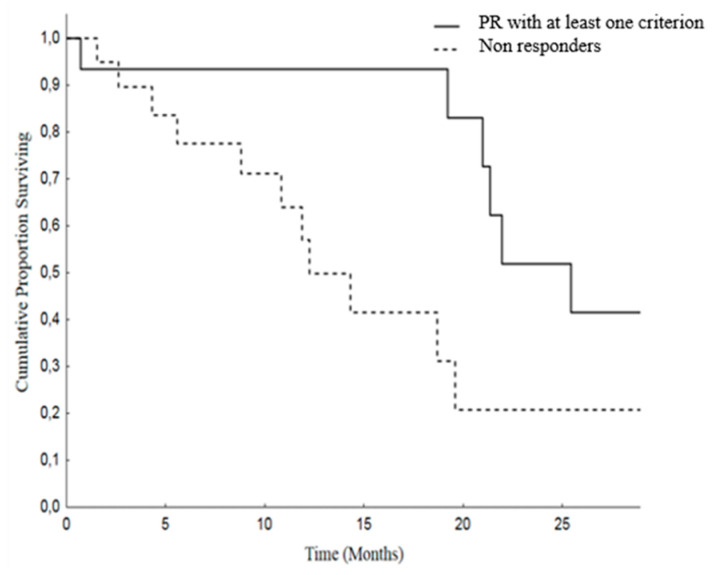
Overall Survival (OS) in patients with PR with at least one criterion [median 25.4 months (19.3–31.4)] versus no response disease [median 12.2 months (8.2–16.2)], *p* = 0.019.

**Figure 3 cancers-12-01395-f003:**
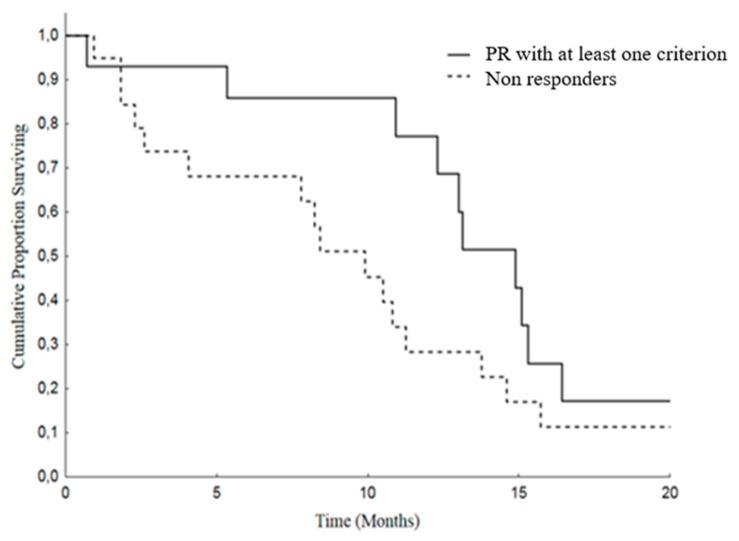
Progression Free Survival (PFS) in patients with PR with at least one criterion [median 14.9 months (11.8–18.0)] versus no response [median 9.9 months (6.8–12.9)], *p* = 0.093.

**Figure 4 cancers-12-01395-f004:**
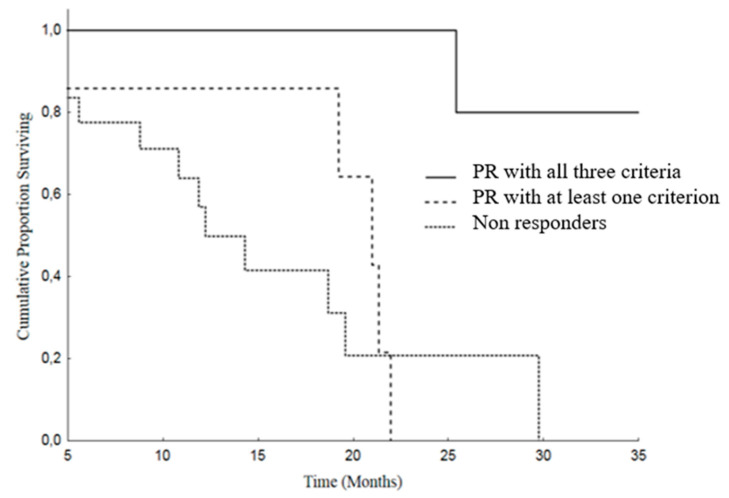
Overall Survival (OS) in patients with PR according to all three criteria [median 37.7 months (19.8–55.6)] versus PR in one or two criterion [median 21 months (17.3–24.6)] versus SD or PD (in all 3 criteria) [median 12.2 months (8.2–16.2)], *p* = 0.005.

**Figure 5 cancers-12-01395-f005:**
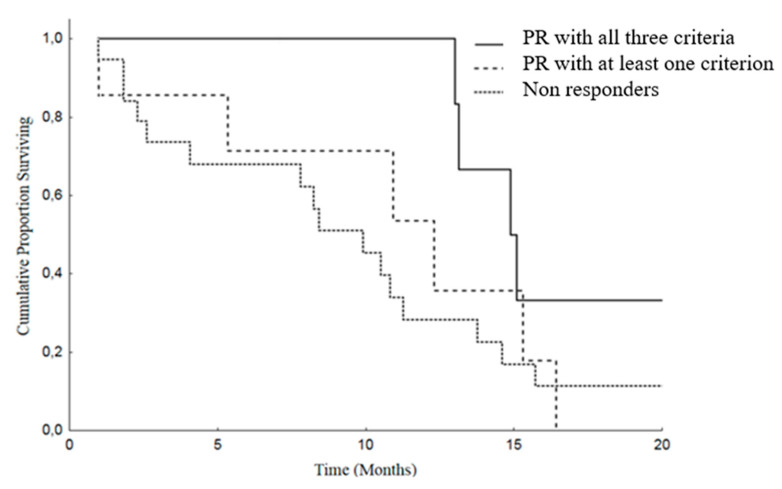
Progression Free Survival (PFS) in patients with PR according to all the three criteria [median 14.9 months (12.5–17.2)] versus PR in one or two criteria [median 12.3 months (4.7–19.8)] versus SD or PD (in all 3 criteria)[median 9.9 months (6.8–12.9)], *p* = 0.107.

**Figure 6 cancers-12-01395-f006:**
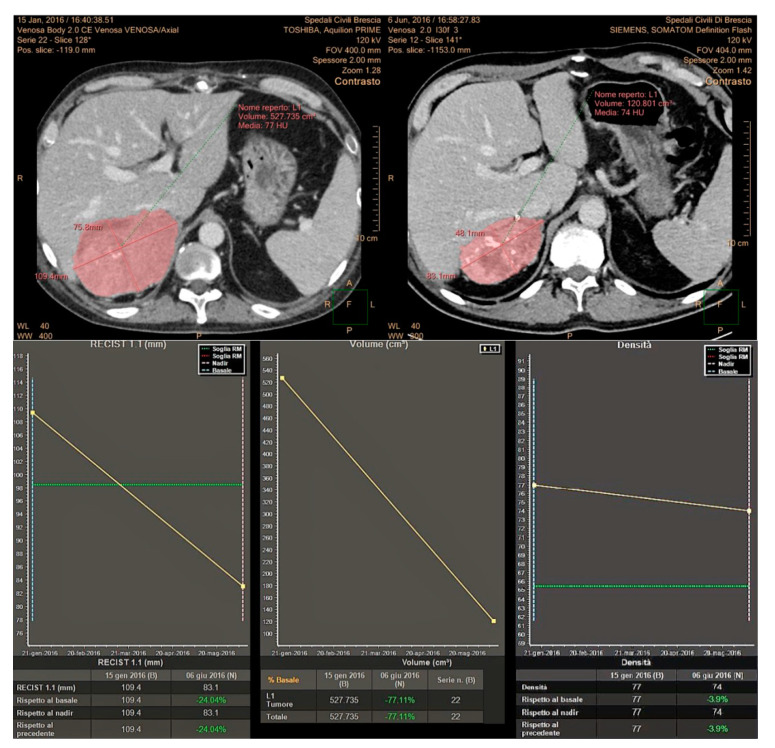
Concordance of the three response criteria in the assessment of a primary metastatic ACC, with planar dimensions changes (RECIST 1.1), tumor volume and tumor size/attenuation (according to Choi).

**Figure 7 cancers-12-01395-f007:**
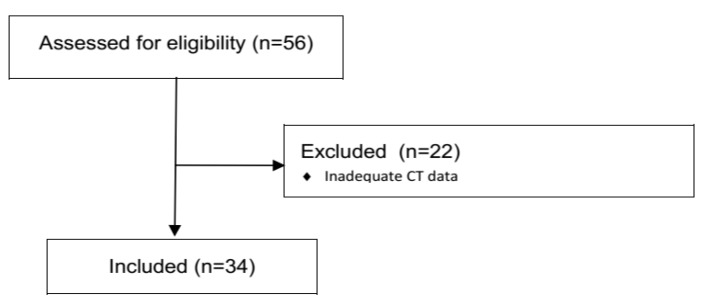
CONSORT Diagram describing the population of the study.

**Figure 8 cancers-12-01395-f008:**
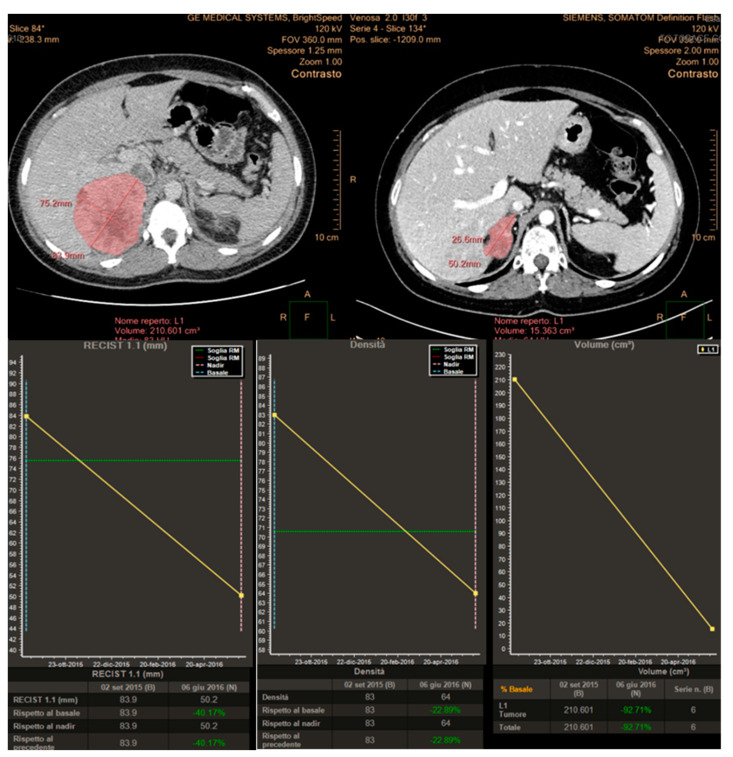
Analysis of variations of tumor characteristics (longest diameters for RECIST 1.1, attenuation changes for Choi and Volume changes) obtained after semi-automatic tumor segmentation and displayed in a chart using the “Multimodality Tumor Tracking” Software (IntelliSpace Portal-Philips).

**Table 1 cancers-12-01395-t001:** Patients demographics and clinical characteristics.

Age	Median	46.3
	Range	16–71
Gender	Female	24 (70.6%)
	Male	10 (29.4%)
Secretory Status	Yes	18 (52.9%)
	No	16 (47.1%)
	Cortisol	15 (83.3%)
	Others	3 (16.6%)
Previous Surgery	Yes	16 (47.1%)
	No	18 (52.9%)
Ensat Staging (at the EDP start)	III	4 (11.8%)
	IV A	16 (47.1%)
	IV B	6 (17.6%)
	IV C	8 (23.5%)
Ki67 at diagnosis	<20	7 (20.6%)
	≥20	9 (26.5%)
	Missing	18 (52.9%)
Death	Yes	20 (58.8%)
	No	14 (41.8%)

**Table 2 cancers-12-01395-t002:** Progression free survival (PFS) univariate analysis according to RECIST 1.1, Choi and volume staging criteria.

CT Response Criteria	Response Category	*n*	PFS (Median)	HR	95% IC Lower Limit–Upper Limit	*p*
RECIST 1.1	PR ^1^	10	14.9	0.329	0.099–1.086	0.125
	SD ^2^	18	9.9	0.799	0.309–2.066	
	PD ^3^	6	1.8			
Choi	PR	14	14.9	0.338	0.120–0.949	0.096
	SD	13	9.9	0.610	0.225–1.656	
	PD	7	10.8			
Tumor Volume	PR	10	14.9	0.313	0.104–0.939	0.087
	SD	17	9.9	0.708	0.282–1.775	
	PD	7	2.6			

^1^ PR, partial response; ^2^ SD stable or ^3^ PD progressive disease.

**Table 3 cancers-12-01395-t003:** Overall, Survival (OS) univariate analysis according to RECIST, Choi and volume staging criteria.

CT Response Criteria	Response Category	*n*	0S (Median)	HR	95% IC Lower Limit–Upper Limit	*p*
RECIST 1.1	PR ^1^	10	37.7	0.122	0.022–0.672	0.006
	SD ^2^	18	18.7	1.063	0.310–3.642	
	PD ^3^	6	14.3			
Choi	PR	14	25.4	0.263	0.264–2.617	0.053
	SD	13	18.7	0.831	0.750–0.925	
	PD	7	14.3			
Tumor Volume	PR	10	37.7	0.146	0.034–0.620	0.001
	SD	17	18.7	0.733	0.235–2.285	
	PD	7	14.3			

^1^ PR, partial response; ^2^ SD stable or ^3^ PD progressive disease.

**Table 4 cancers-12-01395-t004:** Summary of response assessed by the RECIST 1.1, Choi and Tumor Volume criteria.

Response	RECIST 1.1	CHOI	Tumor Volume
**Complete Response (CR)**	Disappearance of all target lesions or lymph nodes <10 mm in the short axis	Disappearance of all target lesions	Disappearance of all target lesions
**Partial Remission (PR)**	>30% decrease in sum of longest diameters (SLD) of target lesions	≥10% decrease in tumor size or ≥15% decrease in tumor attenuation at computed tomography (CT); no new lesions	>65% decrease in tumor volume
**Progressive Disease (PD)**	>20% increase in SLD of target lesions with an absolute increase of ≥5 mm; new lesions	≥10% increase in SLD of lesions; does not meet the criteria for partial response by virtue of tumor attenuation, new intratumoral nodules, or an increase in the size of the existing intratumoral nodules	>73% increase in tumor volume
**Stable Disease (SD)**	None of the above	None of the above	None of the above

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
