# Peer review of "Adrenocortical Carcinoma and CT Assessment of Therapy Response: The Value of Combining Multiple Criteria"

_cancers, 2020, doi:10.3390/cancers12061395_

Round 1

Reviewer 1 Report

The study from Ambrosini et al. explores the value of combining multiple criteria in defining the response to current standard treatment (EDP-M) in patients with advanced adrenocortical cancer (ACC).  The number of patients included in the study (34 stage II-IV) is not high but considering the rarity of these cancers it is not negligible (particularly in a mono-centric study). This might reflect the high expertise in the field of some components of the study-group).

The study is interesting, well written and might have important clinical implications although it might be improved.    

Comments

  • I would suggest to add also a separate analysis of SD and PD.
  • I would suggest to add a comparison between the three individual systems (Recist vs Choi vs volumetric) reporting the accuracy of these systems in the evaluated population
  • I would suggest to add a comparison between different combinations of systems (i.e. Recist+ Choi vs Recist alone etc.)
  • I would suggest to add a subgroup analysis to explore whether the accuracy of the evaluated systems change in different subgroups (i.e. Stage III vs IV; High vs low Ki 67; high vs low tumor burden)
  • In the abstract the three methodologies are listened in a different order in line 24 and in line 27, I suggest to follow always the same order.
  • Line 102: “IVC” should be “IV C”
  • Line 118: “partial response” should be indicated as “PR” because was previously indicated with this acronym, please revise all acronyms.

Author Response

  1. I would suggest to add also a separate analysis of SD and PD.

We performed the requested separate analyses and added them to the manuscript (page 4-5, Line 139-148, tab 2-3).

  1. I would suggest adding a comparison between the three individual systems (RECIST vs Choi vs volumetric) reporting the accuracy of these systems in the evaluated population.

A Receiver operating characteristic curve (ROC) analysis was conducted to assess the global accuracy of disease response by the 3 criteria for predicting PFS and OS, categorized at the median value.

All the three response criteria provided a similar moderate accuracy for predicting PFS (area under curve (AUC) being 0.595, 0,668 and 0.654 for RECIST, CHOI and Volume criteria, respectively.  The corresponding AUCs for OS were: 0.717, 0.704, 0.717, respectively.

This analysis was added in the patient and methods section (page 13, lines 350-352) and in the Results section (page 5, lines 149-152)

Criteria (OS)

AUC

RECIST

0.717

Choi

0.704

Volume

0.717

Criteria (PFS)

AUC

RECIST 1.1

0.595

Choi

0.668

Volume

0.654

.

  1. I would suggest to add a comparison between different combinations of systems (i.e. RECIST + Choi vs RECIST alone etc.)

We performed analysis to explore whether discordant results of CHOI and volume criteria would have a prognostic significance with respect to PR assessed by RECIST, either in term of PFS or OS.

Comparison between Partial Response in RECIST vs Choi (responder in Choi and not in RECIST)

OS p 0.006

Median (months)

Inf

Sup

0 no PR

12.2

8.2

16.2

1 PR according to RECIST

37.7

19.3

56.2

2 PR Choi no PR RECIST

21

4.1

37.8

Comparison between Partial Response in RECIST vs Volume (PR in volume and not in RECIST)

OS p 0.006

Median (months)

Inf

Sup

0 no PR

14.3

6.5

22.0

1 PR according to RECIST

37.7

19.3

56.2

2 PR volume  no PR RECIST

19.2

-

-

PROGRESSION FREE SURVIVAL

Comparison between Partial Response in RECIST vs Choi (PR in Choi and not in RECIST)

PFS p 0.130

Median (months)

Inf

Sup

0 no RP

9.9

6.8

12.9

1 PR according to RECIST

14.9

10.3

19.4

2 PR CHOI no PR RECIST

12.3

0

27.2

Comparison between Partial Response in RECIST vs Volume (PR in Volume and not in RECIST)

PFS p 0.102

Median (months)

Inf

Sup

0 no RP

9.9

6.4

13.3

1 PR in RECIST

14.9

10.3

19.4

2 PR discordant (PRVolume, PD-SD in Recist)

15.3

-

-

  1. I would suggest to add a subgroup analysis to explore whether the accuracy of the evaluated systems change in different subgroups (i.e. Stage III vs IV; High vs low Ki 67; high vs low tumor burden)

Unfortunately, these interesting subgroup analyses could not be done due to the low number of patients included in the study.

Reviewer 2 Report

The article entitled “Adrenocortical Carcinoma and CT assessment of therapy response: the value of combining multiple criteria” by Roberta Ambrosini et all. is a very interesting prospective study that evaluate the disease response of advanced adrenocortical carcinomas treated with EDP-M using RECIST 1.1, Choi and volume criteria. The authors analyzed in 34 patients the response assessment with each criterion with progression free and overall survival; they also evaluate the combination of the 3 criteria on patients outcomes.

They found a better prognosis in the patients in whom the response was agreed upon by 3 criteria; a better survival was found also in patients that the response was identified by 1 or 3 criteria but in this group, there wasn’t a significant advantage in term of progression free survival.

These results are very interesting in term of clinical implication.

This approach could allow patients to be stratified into different subgroups and identify potential patients who could benefit from surgical treatment of metastases from those who do not respond to medical treatment and are in metastatic progression.

I recommend only to move the part of the “Patients and Methods” (line 264) after the chapter “introduction” and before the chapter “Results” to the line 95.

This small modification, would allow a better understanding of the patient population analyzed.

Author Response

  • I recommend only to move the part of the “Patients and Methods” (line 264) after the chapter “introduction” and before the chapter “Results” to the line 95. This small modification, would allow a better understanding of the patient population analyzed

We would like to thank-you the reviewer for appreciating our paper, we have adopted the template provided by Cancers, that probably could not be changed.

Reviewer 3 Report

In the study are evaluateddiseaseresponse of advanced/metastatic ACC patientsuniformlytreated with the standard EDP-M regimenusing RECIST 1.1, CHOI and volume criteria.

The studyaimed to correlate responseassessment with eachcriterion with progression free survival and overallsurvival and to evaluate the combination of threecriteriacoludoffer more helpful information on patientsoutcome

Adrenocortical carcinoma is a rare neoplasm,this explains the small number of patients enrolled in the period considered (2013-2019). The study is defined as a prospective observational, in the paragraph of statistical methods the estimate of the sample size is not reported as provided for this type of study. Furthermore, no evaluation information is published by the Ethics Committee, which assesses the adequacy of the sample size

34 patients were evaluated with the three criteria, the percentage of agreement in terms of partial response for all three criteria is equal to 23.5% (8 patients out of 34). The agreement percentage for at least one criterion is 44.1% (15 patients out of 34).

Considering a proportion success of 40%, test significantlevel a=0.05, 2 sidedtest ,nullhypothesisagreement 10% alternative agreement 24%, with 34 patients a power of about 12% is obtained

Considering a proportion success of 40%, test significant level a=0.05, 2 sided test , nullhypothesisagreement 10% alternative agreement 44%, with 34 patients a power of about 12% is obtained

The study is obviously underpowered for the primary objective. Although the results are very interesting also with regard to survival data, it is appropriate to consider them only exploratory

Author Response

  • The study is obviously underpowered for the primary objective. Although the results are very interesting also with regard to survival data, it is appropriate to consider them only exploratory
  • We absolutely agree with the referee observations. We specify in the methods the explorative nature of the study (page 5, line 343-346) and added in the discussion the limited power of our series including the low potency to detect the difference in absolute response rate between RECIST and at least one of the 3 criteria evaluated (page 10, line 269-276).

  • Methods and Reaserch design must be improved

We followed the referee suggestions to improve the methods section
